# FuseAnyPart: Diffusion-Driven Facial Parts Swapping via Multiple Reference Images

**Zheng Yu** [*][†]
Shanghai Jiao Tong University & Alibaba Group
cs-yuzheng@sjtu.edu.cn

**Yaohua Wang** [*][‡]
Alibaba Group
xiachen.wyh@alibaba-inc.com

**Siying Cui** [†]
Peking University & Alibaba Group
cuisiying.csy@alibaba-inc.com

**Aixi Zhang**
Alibaba Group
aixi.zhax@alibaba-inc.com

**Wei-Long Zheng**
Shanghai Jiao Tong University
weilong@sjtu.edu.cn

**Senzhang Wang**
Central South University
szwang@csu.edu.cn

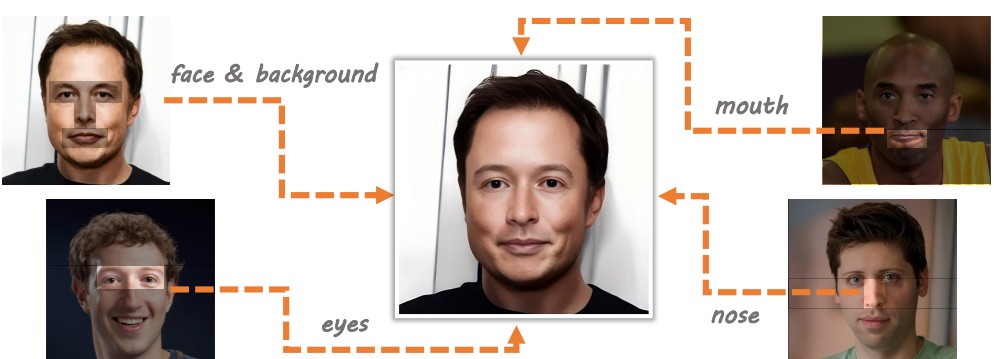

Figure 1: Results of facial parts swapping using the proposed FuseAnyPart at $512 \times 512$ resolution. The swapped face (central image) is generated by fusing the original face (top-left image) with three facial part reference images (bottom-left, top-right, bottom-right). Notably, FuseAnyPart can seamlessly blend facial parts from multiple reference images with significant differences in appearance, producing high-fidelity and natural-looking swapped faces.

## Abstract

Facial parts swapping aims to selectively transfer regions of interest from the source image onto the target image while maintaining the rest of the target image unchanged. Most studies on face swapping designed specifically for full-face swapping, are either unable or significantly limited when it comes to swapping individual facial parts, which hinders fine-grained and customized character designs. However, designing such an approach specifically for facial parts swapping is challenged by a reasonable multiple reference feature fusion, which needs to be both efficient and effective. To overcome this challenge, FuseAnyPart is proposed to facilitate the seamless "fuse-any-part" customization of the face. In FuseAnyPart,

[*]Equal contribution.

[†]Work done during the internship at Alibaba Group.

[‡]Corresponding author.

38th Conference on Neural Information Processing Systems (NeurIPS 2024).

facial parts from different people are assembled into a complete face in latent space within the Mask-based Fusion Module. Subsequently, the consolidated feature is dispatched to the Addition-based Injection Module for fusion within the UNet of the diffusion model to create novel characters. Extensive experiments qualitatively and quantitatively validate the superiority and robustness of FuseAnyPart. Source codes are available at `https://github.com/Thomas-wyh/FuseAnyPart`.

# 1   Introduction

Imagine a person who possesses the face of Elon Musk, the eyes of Mark Zuckerberg, the nose of Sam Altman, and the mouth of Kobe Bryant. What would the visual composite of such a person look like? Currently, this dream can be realized through the facial parts swapping technology as shown in Fig. 1. Different from traditional face swapping, which is coarse and typically replaces the entire face at once, facial parts swapping aims to transfer the individual facial components, e.g., nose, mouth or eyes from varied sources onto the target image while maintaining the rest of the target image unchanged. A growing interest in facial parts swapping technology has emerged due to its broad applications such as innovative character creation, popular entertainment, privacy protection and beyond [29, 20].

Most of the studies so far have primarily focused on full-face swapping, and can be roughly divided into GAN-based and diffusion-based approaches. The GAN-based methods [7, 34, 27, 15, 12] usually perform face swapping by extracting the identity feature from the source images and then injecting them into generative adversarial networks [6]. Nevertheless, the GAN-based techniques may not succeed in completely transferring the identity features, especially when there is a significant difference in shape between the source and the target. In addition, the GAN-based methods often involve an array of losses about image fidelity, identity, and facial attributes to guide the training, which increases the complexity of the training process. On the other hand, diffusion-based models [23, 24, 36, 13, 31, 26, 33] have demonstrated a powerful capability in generating images with high resolution and complex scenes. Some efforts try to swap face [38, 18] through diffusion models, achieving pleasant results.

However, the aforementioned methods, designed for full-face swapping, are either unable or significantly limited when it comes to swapping individual facial parts. If one needs to replace a facial part, it is only possible to swap the entire face, rather than independently swapping one or several facial parts individually, let alone the facial parts from different individuals. This high degree of coupling poses an inconvenience for users who seek more fine-grained and customized designs. Therefore, the focus on face-swapping shifts from an identity-centric to an attribute-level perspective.

The primary challenge in facial parts swapping lies in the fusion mechanism. Popular face-swapping techniques [15, 38, 36, 18] perform the fusion of source and target images in the latent space for harmonious generated images. Therefore, the feature fusion mechanism becomes critical in affecting the quality of the generated images. In the facial parts swapping task, the number of source images increases from one to multiple, further complicating the fusion process and making this issue more prominent. Previous methods [36, 13] utilize adapters implemented by a cross-attention mechanism to fuse reference information into the UNet of diffusion models. However, as the cross-attention is initially designed for multi-modal tasks [24], like text and image, it may be sub-optimal for facial parts swapping due to the difficulty in aligning fine-grained facial region features. What is more, the inclusion of multiple references increases computational needs, thus efficient fusion is essential.

To tackle these challenges, an innovative diffusion-driven approach dubbed FuseAnyPart is proposed to facilitate the seamless "fuse-any-part" customization of faces. In FuseAnyPart, a facial image is initially detected by an open-set detector to derive its various facial part masks. Then an image encoder extracts the facial part features based on the facial image and the aforementioned masks. Subsequently, these facial part features are assembled according to the masks within the Mask-based Fusion Module included in FuseAnyPart to generate a complete face in latent space. After this step, the cohesive feature is forwarded to the Addition-based Injection Module proposed by FuseAnyPart for fusion within the UNet of the diffusion model. The Addition-based Injection Module adds a minimal amount of parameters yet is highly effective in preserving the positional information and fine details of the image features, which demonstrates obvious superiority compared to the conventional cross-attention mechanism. During the training stage, FuseAnyPart is trained by **reconstructing** a

facial image conditioned on different facial parts, inspired by [15, 38]. In the inference stage, facial parts from images of various people can be fed into FuseAnyPart to create a novel character.

Overall, the contributions can be summarized as follows: (1) To the best of our knowledge, Fuse-AnyPart is the first diffusion-based work specifically designed for facial parts swapping, which is capable of simultaneous, multi-source and fine-grained facial parts swapping. (2) The proposed Masked-based Fusion Module in FuseAnyPart allows dynamically aggregating specific parts from different faces in latent space. Then, the Addition-based Injection Module of FuseAnyPart injects this conditional information into UNet, which is more effective and efficient than the conventional cross-attention-based adapter methods. (3) Extensive experiments validate the superiority of Fuse-AnyPart. Ablation studies confirm the soundness of our design choices and the robustness of our proposed approach.

## 2 Related Work

**Image Generation with Multiple References.** InstantBooth [26], InstanID [31], and IDAdapter [3] use the average feature of all reference images, which contributes to improving generation quality. And photoMaker [13] generates an ID embedding by stacking embeddings from multiple ID reference images, which results in improved ID representation. Moreover, it can create a mixed ID embedding by controlling the proportion of identity images within the input reference image collection. Although the aforementioned methods introduce a multiple reference image mechanism, the role of these reference images is similar to that of providing a single reference image, affecting only the generation of a specific subject. FastComposer [33], on the other hand, achieves the generation of multiple subjects by injecting different reference image features into distinct word embeddings. Currently, image generation using multiple reference images remains an area ripe for exploration, such as generating human faces with multiple inputs.

**Facial Parts Swapping.** In recent years, region-controllable face swapping has emerged as a fascinating subfield within the broader domain of facial manipulation and generation. This technology enables precise control over specific regions of a face in an image, allowing the exchange or modification of features such as the eyes, nose, or mouth, while maintaining the integrity of the original image's context. The E4S model [15] achieves precise editing results by manipulating masks of specific regions, such as the eyes or lips, using a reference image as a guide. Meanwhile, Diffswap [38] is a technique that selectively determines the regions to swap by constructing masks that cover varying facial areas. Although these methods are capable of transferring specific facial regions from a source image to a target image, the results often exhibit unnatural boundaries. Furthermore, to achieve the replacement of facial features from multiple reference images onto a single face, multiple iterations are typically required, complicating the process. This iterative approach can be time-consuming and may not consistently produce seamless, natural-looking results, indicating that there is still room for improvement in the field of region-controllable face swapping technology.

## 3 Method

### 3.1 Preliminary

**StableDiffusion.** Our model is based on the StableDiffusion [24] model, which progresses the diffusion process in low-dimensional latent space with a pre-trained autoencoder. Using a latent representation, StableDiffusion can maintain the essential features and structure of the data while requiring fewer steps and less time to generate high-quality samples. First, the variational autoencoder compresses the input image $x$ to a latent representation $z_0$, which is gradually added Gaussian noise with a fixed Markov chain of $T$ steps. Let $z_t = \alpha_t z_0 + \sigma_t \epsilon$ be the noised data at the t-th timestep, where $\alpha_t, \sigma_t$ are predefined functions of $t$ and $\epsilon \in \mathcal{N}(0, I)$, UNet $\epsilon_\theta$ is responsible for the denoising process by predicting the noise $\epsilon$. The denoising process can be conditioned by the additional condition $C$. The training objective is to minimize the ELBO of the denoising process, which is defined as:

$$\mathcal{L} = \mathbb{E}_{z_t, t, C, \epsilon}[||\epsilon - \epsilon_\theta(z_t, t, C)||_2^2]. \tag{1}$$

During inference time, UNet gradually predicts the noise $\epsilon_\theta(x_t, t)$ and recovers the initial latent representation $z_0$ from random noise $z_T \in \mathcal{N}(0, I)$. Finally, the image is generated by mapping $z_0$ back to pixel space with the variational autoencoder.

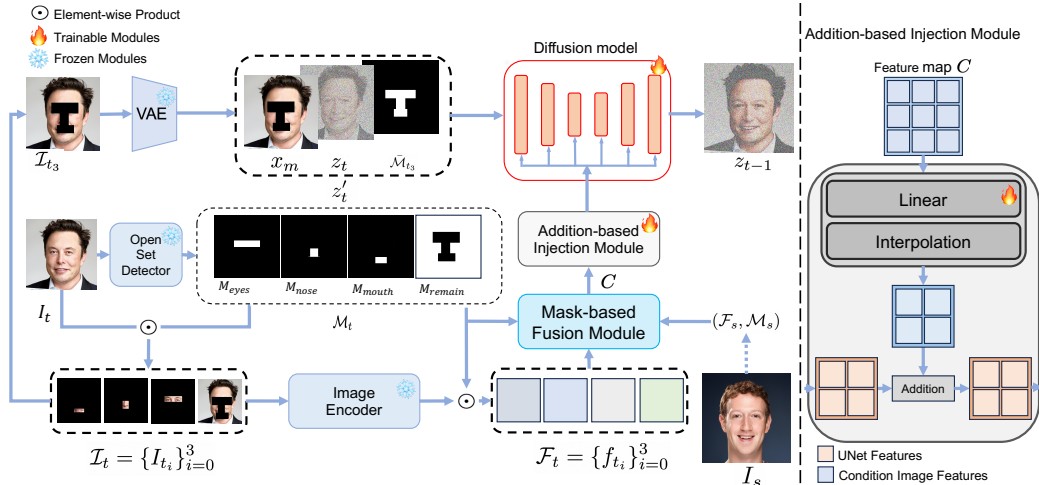

Figure 2: Illustration of FuseAnyPart. The process begins with an open-set detector identifying a facial image to obtain various facial part masks. Following this, an image encoder uses these masks and the facial image to derive the corresponding facial part feature. These facial part features and masks are then fed into the Mask-based Fusion Module to piece together a complete face in latent space. Subsequently, the consolidated feature is dispatched to the Addition-based Injection Module for fusion within the UNet of the diffusion model.

**Image Prompt Adapter.** Image prompt adapter [36] is an innovative approach to incorporate image features into the generation process without model fine-tuning for each new concept. This approach addresses the challenge faced by previous methods, which struggled to effectively extract and utilize detailed image features from image prompts. Similarly to text prompts, image prompts can also condition the generative process through the cross-attention mechanism. Specifically, a decoupled cross-attention strategy is employed in which an additional cross-attention layer is added to every original cross-attention layer to inject image features. The output of this new cross-attention layer can be articulated as

$$Z^{new} = \textbf{Attention}(Q, K, V) + \lambda \cdot \textbf{Attention}(Q, K^i, V^i), \tag{2}$$

where $\lambda$ is weight factor, $\textbf{Attention}(Q, K^i, V^i)$ is cross-attention of the new added cross-attention layer, $K^i = c_i W_k^i$ and $V^i = c_i W_v^i$ are key and values matrices of the corresponding operation. $c_i$ are the image features, and $W_k^i$ and $W_v^i$ are the relevant weight matrices.

## 3.2 Overview

The goal of facial parts swapping is to selectively transfer regions of interest such as the eyebrows, eyes, nose, or mouth from the source image onto the target image while maintaining the rest of the image unchanged. Some methods [15, 8] swap region feature obtained by mask pooling to facilitate facial parts swapping and use masks as structure guidance to maintain detail and coherence in the generated results. However, the applicability of this approach is limited in advanced StableDiffusion models utilizing cross-attention mechanisms, due to the difficulty in aligning fine-grained facial region features with the latent features in UNet. Consequently, it is necessary to fine-tune the entire SD model with a significant volume of data. Additionally, there is a notable paucity of methodologies that permit the incorporation of multiple source images to selectively transfer features from sources onto a target image in one step.

To address these problems, we map a face image into multiple non-overlapping region image features and perform mask-based fusion at the feature map level between the source and target image features. This fusion results in new facial image features, which are then integrated into the generation process through an Addition-based Injection Module.

## 3.3 Facial Feature Decomposition and Aggregation

For simplicity, we consider three regions for swapping including the eyes (including the eyebrows), nose, and mouth. We use an open-set detection model to get region masks $M_{eyes}$, $M_{nose}$, $M_{mouth}$, and the remaining region mask is $M_{remain} = \bar{M}_{eyes} \odot \bar{M}_{nose} \odot \bar{M}_{mouth}$. Let $\mathcal{M} = \{M_{eyes}, M_{nose}, M_{mouth}, M_{remain}\}$ be the region masks, a given face image $I$ can be represented as the union of multiple regional images: $\mathcal{I} = \{I_i\}_{i=0}^3$, where $I_i = I \odot \mathcal{M}_i$.

Following most of the previous methods [2, 3, 37, 32], we utilize a pre-trained CLIP [22] image encoder $\phi$ to extract image representations from regional images. Contrary to the preceding work that harnesses the more abstract, global, and high-level features from the last layer, we use the uncompressed feature map from the penultimate layer, which retains greater spatial information and finer details. For a face image $I$, its feature representation can be decomposed into multiple components $\mathcal{F} = \{f_i\}_{i=0}^3$, where $f_i = \phi(I_i)$. Replacement is conducted at the feature map level, where a target image's features $\mathcal{F}_t = \{f_{t_i}\}_{i=0}^3$, with $i = 0, 1, 2$, are replaceable, corresponding respectively to eyes, nose, and mouth. Mathematically, the feature replacement between target image features $\mathcal{F}_t = \{f_{t_i}\}_{i=0}^3$ and source image features $\mathcal{F}_s = \{f_{s_i}\}_{i=0}^3$ is realized in the **Mask-based Fusion Module**, described as follows:

$$f'_{t_i} = \begin{cases} f_{t_i}, & R_i = True \\ f_{t_i} \odot \bar{\mathcal{M}}_{t_i} + \mathbf{G}(f_{s_i} \odot \mathcal{M}_{s_i}, \mathcal{M}_{t_i}, \mathcal{M}_{s_i}), & R_i \neq True \end{cases} \tag{3}$$

where $i \in \{0, 1, 2, 3\}$, $R_i$ indicates whether the region feature $f_{g_i}$ is replaced by $f_{s_i}$. $\mathbf{G}(I, m_1, m_2)$ is an interpolation function that resizes the region covered by mask $m_1$ in image $I$ to fit the region of mask $m_2$. The resulting features $\mathcal{F}'_t = \{f'_{t_i}\}_{i=0}^3$ are aggregated and then fed into Multi-Layer Perceptron (**MLP**) to generate the final condition feature map, which is input into the UNet as $C = \mathbf{MLP}(\sum_{i=0}^3 \mathcal{F}'_{t_i} \odot \mathcal{M}_{t_i})$ providing the conditional information to guide the generative network.

## 3.4 Addition-based Injection

It has been discussed for a long time how to inject image features into the UNet using a cross-attention mechanism, and there are two principal methodologies. One is a direct method that feeds the concatenation of image features and text features into the layers of cross-attention. However, this method can be ineffective when image features are misaligned with textual features in the concatenation process. The other one, which has been widely adopted in numerous works, employs adapter modules with decoupled cross-attention [36]. Nevertheless, the cross mechanism still faces the challenge of inaccurate feature fusion because attention maps may fail to focus appropriately on the correct regions.

Thus, we propose the **Addition-based Injection Module** for integrating image features into the UNet which directly adds fine-grained image features to latent features within the UNet. Specifically, the output of the injected layer is described as follows:

$$Z' = Z + \lambda \cdot \mathbf{Inter}(\mathbf{Linear}(C)), \tag{4}$$

where $Z$ is the latent feature within the UNet, $C$ is the swapped face image feature map that servers as the condition information, $\mathbf{Linear}(\cdot)$ is a linear layer, $\mathbf{Inter}(\cdot)$ is a function which is capable of resizing $C$ to match the dimensions of $Z$ and $\lambda$ is weight factor. It is feasible because the latent space features within the UNet also comprise a feature map that contains positional information, which has a corresponding spatial relationship with the condition feature map $C$. By adding fine-grained image features at their respective locations, we ensure alignment of the newly introduced features with the original latent space features in terms of position. Furthermore, the injection of such image features is not confined exclusively to the cross-attention layers; it can be integrated at any level within the UNet architecture. Compared to the cross-attention mechanism, this method reduces the number of added parameters and computational load while increasing the flexibility and controllability of feature injection. It enables the model to be fine-tuned with less training data, enhancing efficiency without compromising on the richness of the generated details.

## 3.5 Training and Inference

To preserve the regions of the face image that are not subject to replacement, FuseAnyPart follows the practice of [1]. Specifically, We concatenate the latent vector $x_m$, derived from the masked

image $\mathcal{I}_{t_3}$, the associated mask $\bar{\mathcal{M}}_{t_3}$, and the noised latent vector $z_t$ to form a new latent vector $z'_t = \text{Concat}(x_m, \bar{\mathcal{M}}_{t_3}, z_t)$, which is fed into a convolution layer for dimension adjustment. The feature vector $\hat{z}_t = \text{Conv}(z'_t)$ is subsequently introduced into the UNet, serving as the query. Our training objective is similar to the original StableDiffusion model, formulated as:

$$\mathcal{L} = \mathbb{E}_{z_t, t, x_m, \bar{\mathcal{M}}_{t_3}, C, \epsilon}[|||\epsilon - \epsilon_\theta(z_t, t, x_m, \bar{\mathcal{M}}_{t_3}, C)||_2^2]. \tag{5}$$

During the inference phase, our model possesses the capability to transfer facial regions from multiple source images onto a target image. By deconstructing and reassembling facial image features, we can construct mixed facial features, facilitating flexible and controllable facial parts swapping.

## 4  Experiment

**Dataset.** We train our model on the CelebA-HQ [11] dataset. The CelebA-HQ dataset contains 30,000 high-resolution face images of celebrities widely used for face generation and face swapping tasks. This dataset has been pre-processed and aligned, and is available in three different resolutions. In our experiments, we use the $1024 \times 1024$ resolution. Our evaluation set is sampled from the FaceForensics++ [25] dataset, which contains 1,000 videos. We randomly sample 10 frames from each video and obtain 10,000 images. Then we use GPEN [35] for portrait enhancement and crop and align these images by landmarks to the resolution of $512 \times 512$. Additionally, we collected some high-quality face images from the internet intended for qualitative visual results.

**Implementation Details.** Our implementation is based on HuggingFace diffusers [30] library and we use StableDiffusion v1-5 [24] and OpenAI's clip-vit-large-path14 vison model [22]. We train our model on 16 NVIDIA A100 GPUs (80GB) with a batch size of 16 per GPU using the AdamW optimizer [16] with a constant learning rate of 1e-4 and weight decay of 0.01. During training, facial part reference images are randomly sampled from images with the same ID, and the target image is consistent with the face reference image. During the inference stage, we use the DDIM [28] sampler with 50 steps and set $\lambda = 1.0$. Since we do not use a text prompt, we set the text prompt to empty.

### 4.1  Qualitative Comparisons

We collected a series of high-quality celebrity images from the Internet to conduct qualitative experiments. To demonstrate the effectiveness of our approach, we have structured the qualitative experiments into three sets: fuse any part, multiple parts replacement, and multiple parts replacement with reference images in different styles.

#### 4.1.1  Fuse Any Part

With grounding-dino [14], an open-set object detection model, our method is capable of extracting region-specific features of faces based on text, such as "eyes", "nose", and "mouth". Limited by the performance of grounding-dino, the "eyes" include the "eyebrows" in the subsequent chapters. In this paper, "reference image" is used to describe both the source and target images. The face reference image is the target, while the facial part reference image is the source.

We select eyes, nose, and mouth for attribute-level facial parts swapping. For single part replacement, we compare our method with StableDiffusion (SD)[24], IP-Adapter [36], FacePartsSwap [5], E4S [15] and Diffswap [38], and the results for eyes, nose and mouth are presented respectively in Fig. 3, Fig. 8 and Fig. 9. Since SD and IP-Adapter aren't designed for facial parts replacement, we cut out the desired attributes from the source image and overlaid them onto the target image, resulting in a pixel blend image, the inputs are source-target image pairs. By utilizing SD's image-to-image generation function with the denoising strength set to 0.5, we can reconstruct the spliced image. For the IP-Adapter, the spliced image serves as an image prompt, acting as additional information to condition the generation process. FacePartsSwap specifically focuses on exchanging facial parts and E4S and Diffswap are face swapping methods that can utilize different masks during the inference process to achieve partial facial region replacement.

#### 4.1.2  Multiple Parts Replacement

Multi-attribute replacement differs from swapping the entire facial information onto a target image, as it only involves replacing certain attributes within their corresponding regions, and the number of

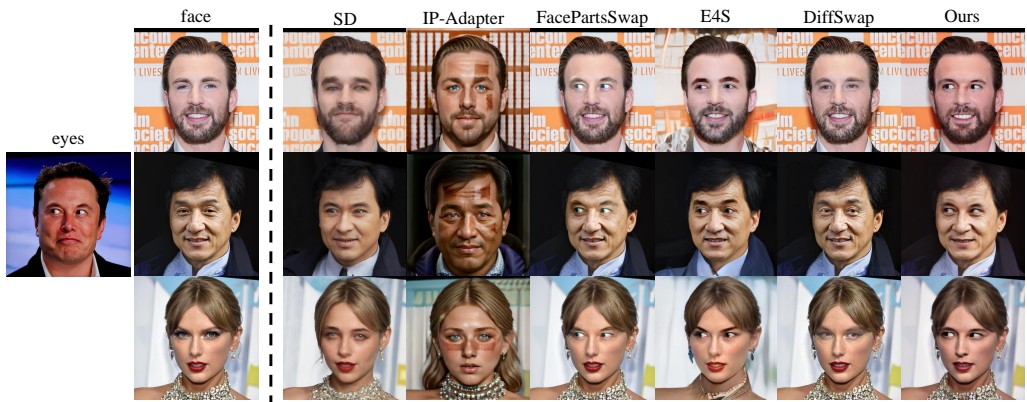

Figure 3: **Qualitative comparison of eyes swapping.** Our method produces high-fidelity results that maintain the consistency of facial features while ensuring a natural appearance.

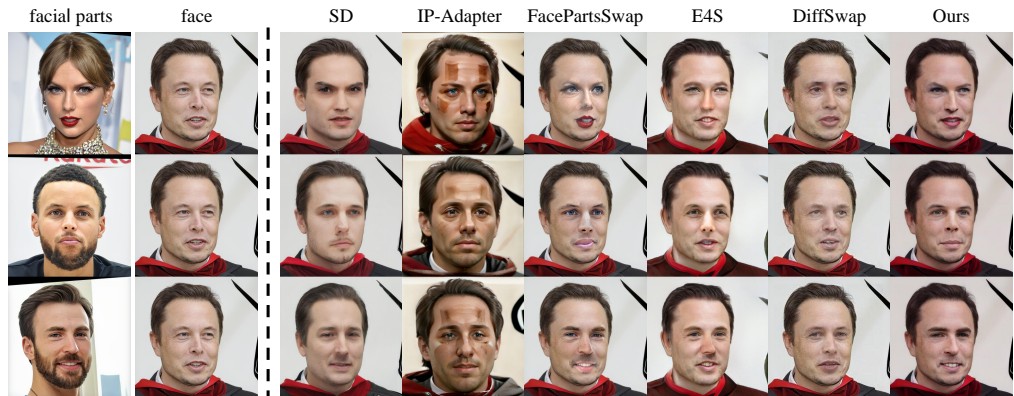

Figure 4: **Qualitative comparison of multiple facial parts swapping with a single reference face.** Our method can naturally replace multiple facial parts of one face with those of another and better preserve both the characteristics and the facial part shape. More results are presented in Fig. 10.

replaced attributes can be arbitrary. We demonstrate the simultaneous replacement of eyes, nose and mouth. Fig. 4 showcases the results of multi-attribute replacement using source-target pairs.

Both the SD and IP-Adapter struggle to maintain the non-replacement areas unchanged, and the similarity of the replaced attributes is not high, highlighting the limitations of pixel space manipulations. While FacePartsSwap and E4S can retain a higher degree of attribute similarity, the replacement results often appear visually inconstant and forced, particularly when there is a significant difference between the source image and the target image, such as in skin tone or facial angle. In contrast, the replacement effect of Diffswap is not pronounced.

Our method outperforms these approaches by offering better consistency across both the replaced attributes and the unaffected areas, leading to a more seamless and natural integration of the replaced features regardless of discrepancies in the reference images. Moreover, Fig. 5 demonstrates the results when each replaced attribute originates from different source images. Our method still significantly surpasses other approaches, as it can combine distinct attribute features and generate natural-looking facial photos. This superior performance indicates that our method effectively extracts and integrates the characteristics of individual attributes, even when dealing with varied sources. It reinforces our method's flexibility and robustness in handling complex face manipulation scenarios where each facial feature may require a different treatment based on its unique reference image.

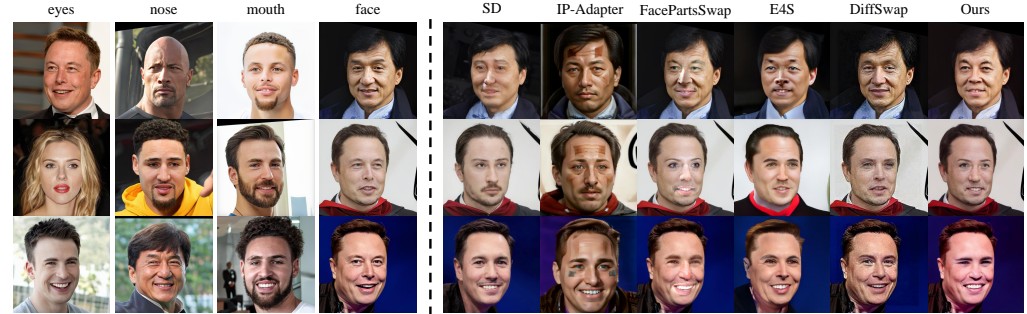

| eyes | nose | mouth | face | | SD | IP-Adapter | FacePartsSwap | E4S | DiffSwap | Ours |

Figure 5: **Qualitative comparison of multi swapping with multiple reference faces.** Our method remains robust to different appearances of various reference facial parts.

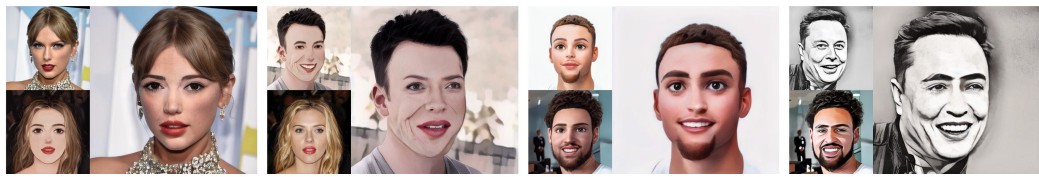

Figure 6: **Facial parts swapping on images with different styles.** As an extended application, FuseAnyPart can use the facial parts of reference images with different styles to generate harmonious faces without changing the style of the target face. We show that the features of the facial parts and the image style can be well decoupled.

### 4.1.3 Fusion Across Different Styles

Fig. 6 shows the results of our method performing multi-attribute replacements on images spanning various styles. This is a limitation often encountered in many face swapping methods, as they typically rely on facial segmentation models which are bound by the constraints of their training data and tend to have poor generalization on unseen data. Benefiting from the open-set detection model, our method can extract any regional feature from the reference images, which helps our method to generalize well on the data with different distributions.

To get images in different styles, we apply a style modification model [19] to real photographs to generate a series of images in different styles, including cartoon, 3D, sketch, and more. We then use these stylized images as references to perform multi-attribute replacements following the experimental setup described earlier. Despite the style discrepancies between reference images, our method is still able to accurately extract and fuse the targeted features. The generated results maintain the characteristics of the reference images while preserving the style of the original face reference. This demonstrates our method's robust capability to handle diverse styles and perform complex attribute fusion tasks effectively.

### 4.2 Quantitative Comparisons

**Evaluation Metric.** Following common practice, we adopt Fréchet Inception Distance (FID) [10, 21] to evaluate the quality of the generated images. Our method is capable of generating faces with multiple reference images, including a face reference image and three facial part reference images (an eyes image, a nose image and a mouth image). To evaluate the effect of facial part reference images, we propose a metric named **FPSim** (Facial Part Similarity) to measure the similarity between the corresponding facial parts of the generated face and those of reference images, and FPSim-E, FPSim-N and FPSim-M are distinct metrics that respectively measure the similarity of the eyes, nose, and mouth. FPSim is defined as $\frac{f_a \cdot f_b}{\|f_a\|\|f_b\|}$, where $f_a$ and $f_b$ are the attribute-level features of the generated face and the corresponding reference images. To extract attribute-level features, we train three facial attribute-level feature extractor models with ResNet50 [9] and ArcFace loss [4] on the CelebA-HQ dataset for eyes, noses and mouths respectively. To measure the ability to reconstruct

Table 1: **Quantitative Comparisons on FF++.** We report Fréchet inception distance, eye similarity, nose similarity, mouth similarity and Mean Square Error and show that our method achieves SoTA or competitive results compared with existing methods. FacePartsSwap is essentially a cut & paste method, rather than a generative one, and thus has a higher FPSim. Therefore, we only present its results here and do not include it in the quantitative comparisons.

| Methods | FID $\downarrow$ | FPSim-E$\uparrow$ | FPSim-N$\uparrow$ | FPSim-M $\uparrow$ | MSE$\downarrow$ |
|---|---|---|---|---|---|
| StableDiffusion [24] | 18.57 | 0.3080 | 0.2215 | 0.2127 | 1.66 |
| IP-Adapter [36] | 69.35 | 0.2865 | 0.2066 | 0.1886 | 13.72 |
| FacePartsSwap [5] | 44.23 | 0.3269 | 0.2190 | 0.2220 | 24.40 |
| E4S [15] | 30.61 | 0.2764 | **0.4047** | 0.1903 | 3.03 |
| Diffswap [38] | 12.07 | 0.2461 | 0.1967 | 0.1731 | **0.15** |
| Ours | **10.54** | **0.3186** | 0.2234 | **0.2196** | 0.77 |

Table 2: **Quantitative comparison of feature fusion under different ablative configurations.** Both the generation quality and facial part similarity are measured.

| Method | FID$\downarrow$ | FPSim-E$\uparrow$ | FPSim-N$\uparrow$ | FPSim-M $\uparrow$ | MSE$\downarrow$ |
|---|---|---|---|---|---|
| Cross-attention | 15.81 | 0.2542 | 0.1763 | 0.1771 | 1.02 |
| Multiple Cross-attention | 15.32 | 0.2407 | 0.1834 | 0.2063 | 1.94 |
| Cross-attention + Addition-in-Conv | 16.66 | 0.2706 | 0.1897 | 0.1797 | **0.66** |
| Cross-attention + Addition-in-CA | **10.51** | 0.3108 | 0.2128 | 0.2158 | 0.71 |
| Cross-attention + Addition-in-CA + Hierarchy | 28.96 | 0.2808 | 0.2034 | 0.2077 | 1.33 |
| Addition-in-CA (Ours) | 10.54 | **0.3186** | **0.2234** | **0.2196** | 0.77 |

Cross-attention: Using cross-attention to inject conditional features.
Add-in-Conv: Addition within convolutional layers.
Add-in-CA: Addition within cross-attention layers.
Hierarchy: Use of hierarchical features.

faces with reference images from the same ID, we compute the Mean Square Error (MSE) between generated images and reference images.

**Quantitative Comparison.** As indicated in Tab. 1, we compare our method with previous methods on the FaceForensics++ [25] dataset. The results show that our method outperforms previous methods in FID significantly, indicating that we can generate high-fidelity swapped faces and can better preserve naturalness and harmony. Meanwhile, we also achieve comparable results on attribute-level metrics, demonstrating that our method can also keep the characteristics of the swapped facial parts. Notably, we observed a limitation in DiffSwap [38], with its tendency to yield results more resemble the source face rather than the intended target face, as illustrated in Fig. 4. Therefore, when both the source and target faces come from the same ID, DiffSwap achieves a lower MSE (0.15) compared to ours (0.77). Nonetheless, it is more common for the IDs of the source and target faces to differ; in these cases, our method consistently shows superior performance.

## 4.3 Ablation Study

Qualitative comparisons are in Fig. 7 and the quantitative comparison is shown in Tab. 2, where both the generation quality and facial part similarity are measured.

**Cross-attention *vs*. Addition.** To inject conditional features of reference facial parts, we directly add the swapped face image feature to the UNet latent feature instead of using cross-attention. As shown in the first and last rows of Tab. 2, direct addition significantly enhances the swapping performance. We also try to use different cross-attention modules for different facial parts and add their results together to form the latent features of the UNet (denoted by "Multiple Cross-attention"). This method provides only a limited improvement to the model's performance (the 2nd row of Tab. 2 and the 4th column of Fig. 7). Moreover, we try to combine the two fusion methods, adding the swapped face image feature to the output of cross-attention layers (the 2nd row of Tab. 2). Although there is a slight improvement in image fidelity (FID and MSE), the facial part similarity all decreased (FPSim-E, FPSim-N and FPSim-M).

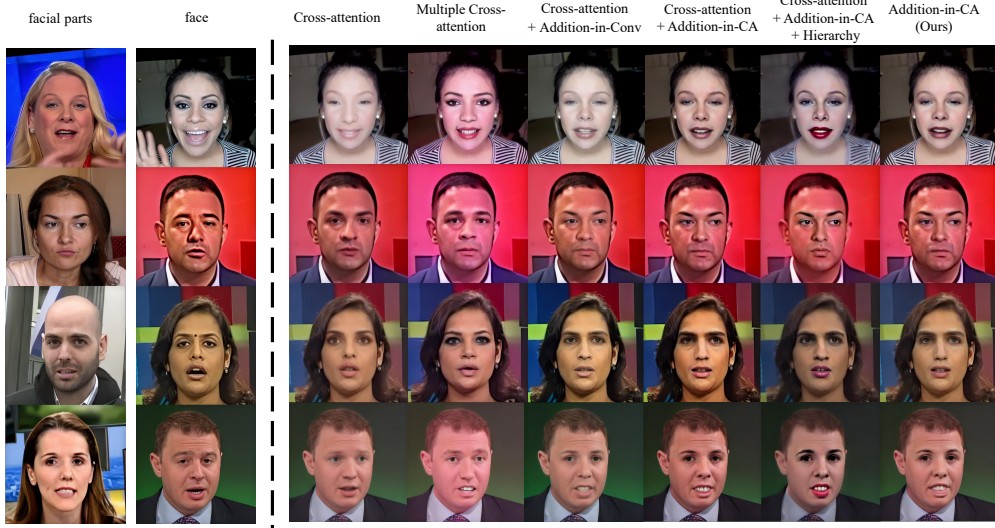

| facial parts | face | Cross-attention | Multiple Cross-attention | Cross-attention + Addition-in-Conv | Cross-attention + Addition-in-CA | Cross-attention + Addition-in-CA + Hierarchy | Addition-in-CA (Ours) |

Figure 7: Qualitative comparison of different ablative settings.

**Feature Injection Across Layers.** We inject the swapped face image feature between the two convolutional layers in each ResNet block of the UNet, rather than in the cross-attention layers. The results in Tab. 2 suggest that the Addition-based Injection Module should be positioned in the cross-attention layers (Row 3 and 4). From the 5th and 6th columns of Fig. 7, we can observe that fusion in the cross-attention layer preserves more details and achieves higher similarity than that in the convolution layers.

**Hierarchical Feature.** Features from the 4th block of the CLIP image encoder and one after every four blocks are extracted and concatenated to form the output of the CLIP image encoder as the hierarchical feature, which contains abundant facial visual information. According to the Row 4 and 5 of Tab. 2, the method is unable to generate high-fidelity and realistic faces with a significant decrease in image quality metrics, which is confirmed by the 7th column in Fig. 7.

## 5  Societal Impacts, Limitations and Conclusion

**Societal Impacts.** The proposed FuseAnyPart is fundamentally harmless. Nevertheless, misuse of it, e.g., applications with copyright issues and racial issues, could have negative effects on society. As a result, we call for a conscientious and ethical implementation of FuseAnyPart.

**Limitations.** While FuseAnyPart demonstrates strong performance in facial parts swapping, it still has some limitations at the current stage. First, while FuseAnyPart performs well on a range of images, there may be challenges with faces that have extreme poses, occlusions, or expressions. Additionally, our method is primarily designed for facial parts swapping and does not directly tackle the challenge of preserving or transforming facial expressions during the process of swapping. FuseAnyPart is based on diffusion models, which typically exhibit high computational complexity due to recursive iterations. Algorithms like Latent Consistency Models (LCM) [17] can accelerate inference by reducing the number of iterations, while techniques such as int8 model quantization can significantly lower computational load. Together, these strategies enhance the speed of FuseAnyPart. Like most generative models, FuseAnyPart relies on high-quality training datasets. The quality of the images can be enhanced using super-resolution methods.

**Conclusion.** This paper proposes FuseAnyPart, a novel diffusion-driven method for facial parts swapping. FuseAnyPart first extracts multiple decomposed features from face images with masks obtained from an open-set detection model. Then parts from different faces are aggregated in latent space with the Mask-based Fusion Module. An injection module injects this conditional information into UNet for fusing effectively. Extensive experiments validate the superiority of FuseAnyPart.

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

# A More qualitative results

As discussed in Sec. 4.1.1, the qualitative comparison of the nose and mouth swaps is presented in Fig. 8 and 9.

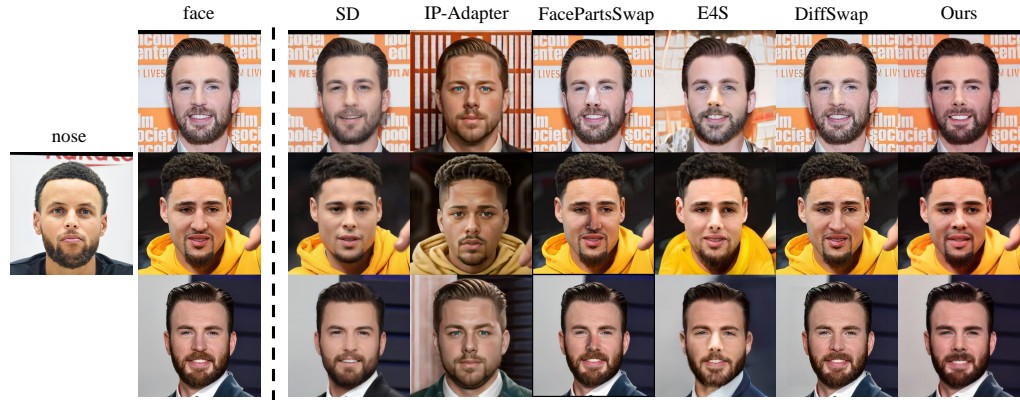

Figure 8: Qualitative comparison of nose swapping.

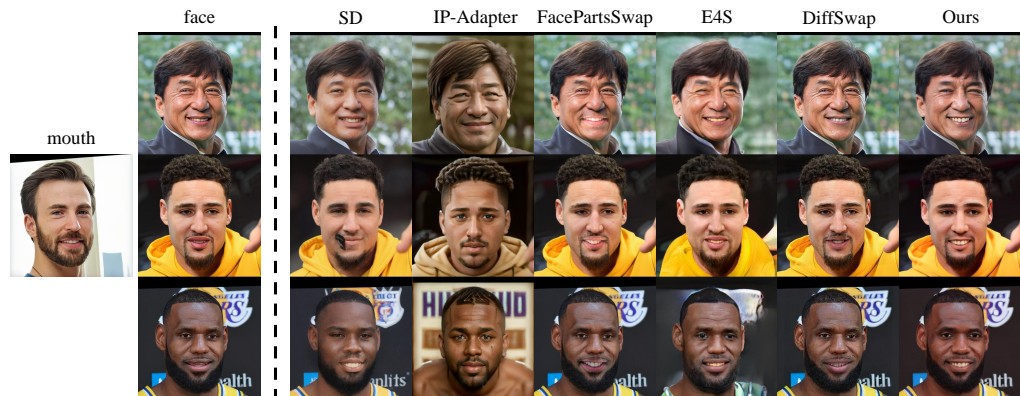

Figure 9: Qualitative comparison of mouth swapping.

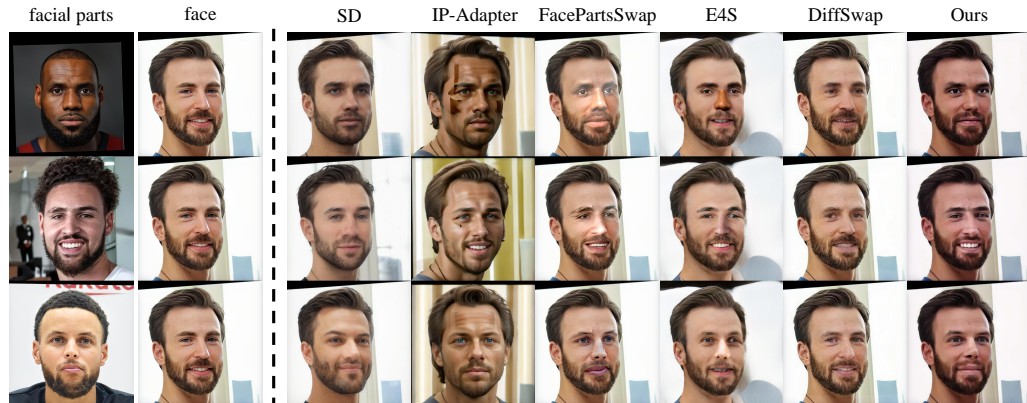

Figure 10: More qualitative comparisons of multiple facial parts swapping. This provides additional examples related to Fig. 4.

As shown in Fig. 11, FuseAnyPart performs well when swapping facial parts from individuals of significantly different racial or age groups.

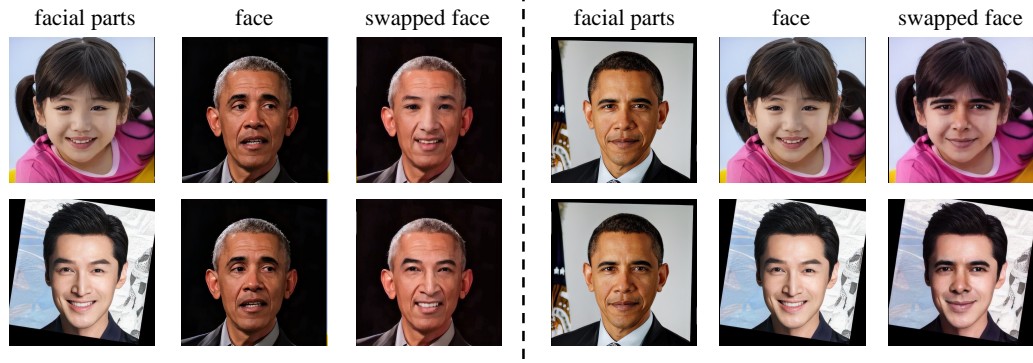

| facial parts | face | swapped face | | facial parts | face | swapped face |

Young Asian to Old Black                    Middle-aged black to Young Asian

Figure 11: Illustrations of facial parts from significantly different racial and age groups. Facial part swapping between source and target images that significantly differ in age and race.

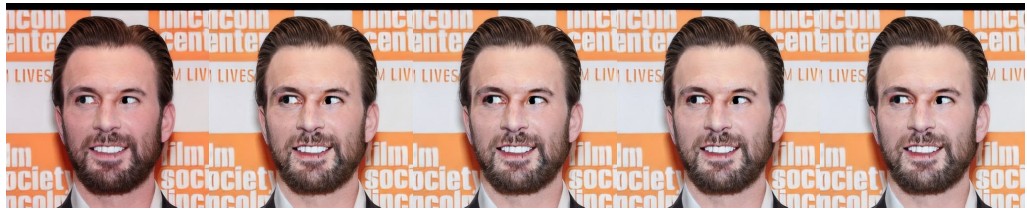

$thres_i = 0 * T$     $thres_i = 0.2 * T$     $thres_i = 0.4 * T$     $thres_i = 0.6 * T$     $thres_i = 0.8 * T$

Figure 12: The skin color change issue can be effectively resolved by replacing the generated skin regions with the inverted latent representations of the original skin color using DDIM inversion in the denoising process. The threshold indicates the number of steps performed above the replacement operation in the denoising process.

FuseAnyPart may encounter issues with color changes in generated images, but this problem can be addressed by replacing the generated skin regions with inverted latent representations of the original skin color using DDIM. The results are presented in Fig. 12.

FuseAnyPart was also qualitatively compared with DiffFace, and the results are shown in Fig. 13. More diverse results of multi swapping with multiple reference faces are presented in Fig. 14.

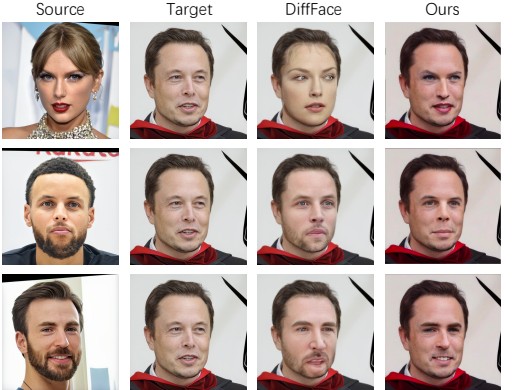 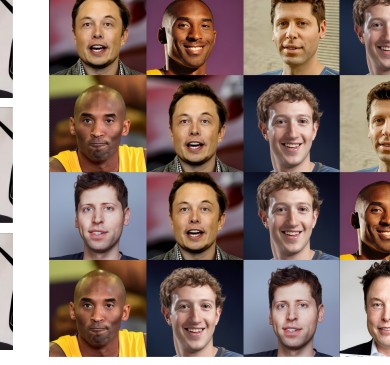 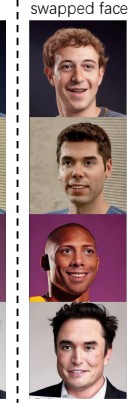

Figure 13: Comparison with DiffFace. DiffFace generates images with local distortions in the eyes and mouth, whereas our method produces cleaner results that are more similar to the source image regarding the facial parts.

Figure 14: Qualitative results of swapping face parts from different sources to a target face.

