# OpenReview forum: "FuseAnyPart: Diffusion-Driven Facial Parts Swapping via Multiple Reference Images"
_NeurIPS.cc/2024/Conference — NeurIPS 2024 spotlight_

### Official Review · Reviewer_mdEG · 2024-07-05

**Soundness:** 3
**Presentation:** 2
**Contribution:** 2
**Rating:** 5
**Confidence:** 4

**Summary:**

This paper introduces a facial parts swapping framework based on diffusion models, named FuseAnyPart. Unlike traditional methods that swap entire faces, FuseAnyPart allows for the swapping of individual facial features. The framework fine-tuned a pre-trained CLIP model to extract features from different facial regions and a stable diffusion model to generate the swapped face. To merge these features, an Addition-based Injection Module is utilized, incorporating the facial region features into the UNet feature map. Extensive experiments validate the effectiveness of the proposed method.

**Strengths:**

1. The paper is easy to follow.
2. Utilizing a diffusion model to achieve region-based face swapping is interesting.
3. Extensive experiments verify the effectiveness of the proposed methods.

**Weaknesses:**

1. The results in Figure 3 show that the skin color also changes when only the eyes are swapped, which the authors did not discuss.
2. The method of feature injection into the SD through addition has already been used in IPAdapter and ControlNet.
3. Since there is no swapped face used for training, is the source face directly used as the target face during training? If so, how is the feature decoupling between different regions achieved during inference?
4. The term `one-step' mentioned in line 65 contradicts the 50 steps of the ddim sampler described in the Implementation Details. The term ‘one-step’ should be used with caution in methods based on diffusion models.

**Questions:**

Fusing multiple facial parts to generate a new face is an interesting idea. However, it is unclear how to evaluate the performances quantitatively. In addition, such generated faces are not visually plausible to me. The FID scores also indicate that the proposed method is not state-of-the-art.

**Limitations:**

The proposed method can only perform facial parts swapping on aligned faces, which limits its applications in practical situations.

---

> ### Author Rebuttal · Authors · 2024-08-06
>
> **[W1. The problem of skin color change.]**
> FuseAnyPart may encounter skin color changes, particularly when there is a significant difference between the skin colors of the source images and the target image.
> This issue arises because the source and target images are fused in the latent space, and the global attention mechanism diffuses the color features.
> However, this of skin color changes issue can be effectively resolved by replacing the generated skin regions with the inverted latent representations of the original skin color using DDIM inversion. This ensures the skin color region remain unchanged.
> The improved results are illustrated in Figure 2 of the newly submitted PDF.
> The pseudocode is as follows.
> ```plaintext
> Algorithm: Merging with Ground Truth Latent
> Input: latents_ref_img, init_noise, pixel_mask
> Output: latents
>
> 1. Initialize thres_i
> 2. If i < thres_i
>    a. Retrieve noise_timestep from scheduler:
>       - noise_timestep <- scheduler.timesteps[i + 1]
>    b. Add noise to the latent of the reference image to obtain ground truth noised latent:
>       - gt_latent <- scheduler.add_noise(latents_ref_img, init_noise, noise_timestep)
>    c. Update latents using pixel mask:
>       - latents <- (1 - pixel_mask) * gt_latent + pixel_mask * latents
> ```
>
> **[W2. Feature injection difference to IPAdapter and ControlNet.]**
> IPAdapter is not an addition-based injection method. It employs a decoupled cross-attention strategy, where an additional cross-attention layer is added to each original cross-attention layer to inject image features. And our method could perform better in facial parts swapping tasks.
> In Section 4.3 of our paper, titled "Cross-attention vs. Addition," we compared our feature injection method with cross-attention. The results indicate that our injection method is superior.
> ControlNet is an addition-based injection method. We have not yet compared it in our experiments. However, for face parts swapping tasks, the majority of the target image remains unchanged, and the features needing injection are only those corresponding to the parts requiring replacement, thus eliminating the need for a complex network structure. Our method involves the addition of only linear layers, whereas ControlNet requires replicating the parameters of UNet. Therefore, our approach has a clear advantage regarding the increase in parameter count. For this task, our method is simple but effective.
>
>
> **[W3. The training strategy.]**
> FuseAnyPart uses reconstruction as a proxy task and does not require swapped paired faces, which draws inspiration from E4S and DiffSwap.
> For instance, the target face image and the source face images, which provide different facial parts, are sampled from different images of the same identity.
> Our paper details the training and inference pipeline in Lines 60-62.
> A visual grounding model is employed to detect bounding boxes for feature decoupling across different regions.
>
>
> **[W4. The term 'one-step'.]**
> The 'one-step' mentioned in Line 65 means that the eyes, mouth, and nose can be swapped in a single inference, unlike traditional inpainting methods, where they are swapped sequentially.
> Therefore, it does not contradict the 50 steps of the DDIM sampler.
> We appreciate the reviewer's careful reading and will revise the paper in the final edition to clarify this point.
> The term 'one-step' in Line 65 will be replaced with 'simultaneous'.
>
>
> **[Q. Evaluate the performances quantitatively.]**
> As described in Section 4.2, three metrics, including Fréchet Inception Distance (FID), Organ Similarity (Osim), and Reconstruction Mean Squared Error (MSE), are adopted to evaluate the performance quantitatively.
> FID is used to evaluate the generated faces' overall image quality and visual realism.
> Osim is used to evaluate how well individual features such as the eyes, nose, or mouth match between the generated image and the sources, focusing on the fidelity of the swap within localized regions.
> The MSE metric measures the pixel-wise accuracy of the reconstructed image against the original. It helps quantify the preservation of the non-swapped parts of the face and the seamless integration of swapped parts.
> Regarding the visual plausibility, we provided some qualitative results in the paper, and additionally, we have included more qualitative results in the newly submitted PDF.
> The results showcase the efficacy of our method in generating plausible faces even when there are significant differences between the source and target images, including differences in age and race.

---

> > ### Comment · Reviewer_mdEG · 2024-08-12
> >
> > Thanks for providing the rebuttal, which addressed my concerns. I have raised my score to borderline accept.

---

### Official Review · Reviewer_jimd · 2024-07-11

**Soundness:** 3
**Presentation:** 3
**Contribution:** 2
**Rating:** 5
**Confidence:** 3

**Summary:**

This paper delves into the strategy of facial parts swapping which are not studied before, their method aims at fusing facial parts from different sources into the overall background/face images. It involves masking facial landmark areas(i.e. eyes, mouth, nose) and fusing with mask-based operation. Finally, the conditions are injected into diffusion models with an Interpolation layer.

**Strengths:**

1. This paper addresses the problem of part-level facial at the feature level which is interesting.
2. The proposed injection method improves the final generation than the traditional cross-attention method.
3. Qualitative and Quantitative experiments have shown the effectiveness of the proposed method.

**Weaknesses:**

1. the evaluation metric(OSim) about whether the face is correctly swapped is not explained in detail, maybe providing more text description is easier for the reader to understand. For example, does the OSim-E input only include the eye image, or is it the entire image with only the eyes unmasked? Additionally, what is the label used for training OSim, is it based on identity?

2. the Addition-based Injection Module is simple, does advanced modules ( for example convolution/ self-attention) lead to inferior performance?

**Questions:**

1. lines 163-166, do you have literature or visualization results to prove 'method can be ineffective when image features are misaligned with textual features'?

2. What is the target/labelled image for training, considering there is no correct swapped reference image? Is it classifier-free guidance?

3. Is the code available?

4. Which version of the VAE model is used in this work? Is there any lightweight but effective pre-trained VAE?

**Limitations:**

this work is interesting for part-based facial editing. But some details are missing such as the evaluation protocol OSim should be explained in detail, as well as the landmark detection network.

---

> ### Author Rebuttal · Authors · 2024-08-06
>
> **[W1. Details about OSim.]**
> Osim measures the similarity between the swapped facial parts (eyes, nose, mouth) in the generated image and those in the reference images.
> For example, we utilized the CelebA-HQ dataset and employed the grounding-dino detection model to identify and extract bounding boxes for the eyes.
> Each cropped eye region (the eye image) was then labeled with the identity (ID) of the corresponding individual.
> Then, we train the ResNet50 with arcface loss, which maximizes the intra-class similarity (among the same IDs) and minimizes the inter-class similarity (across different IDs).
> After training, we use the ResNet50 model to extract feature vectors from the eye regions. Osim is computed using cosine similarity between these feature vectors: $Osim = \frac{f(a)\cdot f(b)}{\Vert f(a) \Vert \Vert f(b) \Vert}$, where f represents the feature extraction function implemented by our trained ResNet50 model and a and b are the input images of the specific organ.
>
> **[W2. Alternatives to the Addition-based Injection Module.]**
> The Addition-based Injection Module beats multiple attention-based baselines in Table 2 and Figure 7.
> Although simple, the Addition-based Injection Module is the most suitable for the facial parts swapping task and validated by extensive experiments.
>
> **[Q1. The  visualization results.]**
> This point refers to the baseline method, IP-Adapter.
> IP-Adapter depends on both text and image prompts.
> If the text prompt is inaccurate or even conflicts with the image prompt, it may have little effect on the results, leading to outcomes that do not align with the text prompt.
> We will clarify this point to ensure it is clear in the final edition.
> Please check the Figure 5 in the newly submitted PDF.
>
> **[Q2. The target image for training.]**
> FuseAnyPart uses reconstruction as a proxy task for training and does not require swapped paired faces, which draws inspiration from E4S and DiffSwap.
> For instance, the target face image and the source face images, which provide different facial parts, are sampled from different images of the same identity.
> Our paper details the training and inference pipeline in Lines 60-62.
>
> **[Q3. Code available.]**
> Once the paper is accepted, the training and inference code, model weights, and essential documentation will be released to the public.
> The authors aim to make a valuable contribution to the NeurIPS community.
>
> **[Q4. The VAE version.]**
> FuseAnyPart is based on the widely used SD-v1.5, and the VAE model utilized in this work is the same as that in SD-v1.5.
> The authors acknowledge that a lightweight yet effective pre-trained VAE may exist, but exploring this is beyond the scope of FuseAnyPart.
>
> **[L. The landmark detection network.]**
> A visual grounding model is used to detect bounding boxes around faces, rather than relying on a landmark detection network, as described in Line 216.

---

> > ### Comment · Reviewer_jimd · 2024-08-12
> >
> > Thank the authors for the rebuttal. I think this response has addressed my concerns. I will keep the scores as it is, borderline accept.

---

### Official Review · Reviewer_Aivf · 2024-07-12

**Soundness:** 2
**Presentation:** 2
**Contribution:** 2
**Rating:** 5
**Confidence:** 4

**Summary:**

This paper explores the partial face swapping problem. Rather than swapping the whole face from A to B, partial face swapping aims to swap some specific area (or organ) of A to B. In this paper, a diffusion based partial face framework is proposed. Besides, two modules are designed to better fuse the extracted feature into the diffusion unet. Facial Feature Decomposition effectively extracts facial semantics and Addition based Injection module integrates the semantics into the diffusion model. Further experiments demonstrate the effectiveness of the propsoed framework.

**Strengths:**

1. The task (partial face swapping) is interesting and more challenging compared to whole face swapping. It needs fine-grained controls and thus worth exploring.

2. The storyline of this paper is simple and clear.

**Weaknesses:**

1. The paper says "the primary challenge in facial parts swapping lies in the fusion mechanism". Could you please detail it? In my point of view, this challenge also exists in conventional face swapping task. Face swapping manipulates the face area of an image while partial face swapping manipulates a smaller area. Previous mask-based face swapping methods first generate faces with the same expression as the target face and then pastes it to the target face according to the mask.

2. Some details are missing in the method:
* What is the $z_T$ (the initial point of the denoising process) used in the training and inference? is it a Gaussian noise or the target image?
* Why the training objective is to reconstrct the target (Eq 5)?
* What is the dimension of The extracted feature $f$? If it is a H*W feature map, how to fed it into MLP?

3. There are two few qualitative results in the article (the main paper and supp). The experiments adopt CelebA datasets which do not have IDs, how to sample from the same ID (Line 207)? I guess it is sampled from the same image?

4. More comparisons with conventional face swapping method (fsgan, simswap, diffFace) should be given.

**Questions:**

See the weaknesses.

**Limitations:**

The author have addressed the limitations.

---

> ### Author Rebuttal · Authors · 2024-08-06
>
> **[W1. The primary challenge lies in the fusion mechanism.]**
> Traditional face-swapping methods first perform face reenactment and then paste it onto the target image in pixel space, as illustrated in FSGAN and DiffFace.
> However, operations in pixel space often result in unnatural images with visible seams, leading to a low success rate in practical applications and producing many poor outcomes as shown in Figure 3 of the newly submitted PDF.
> The current mainstream face-swapping techniques now perform the fusion of source and target images in the latent space, resulting in more harmonious generated images.
> Therefore, the feature fusion mechanism becomes critical in affecting the quality of the generated images.
> In the facial parts swapping task, the number of source images increases from one to multiple, further complicating the fusion process and making this issue more prominent.
> As shown in Table 2 and Figure 1 of this paper, the authors conducted extensive experiments to validate the superiority of the proposed fusion mechanism quantitatively and qualitatively.
> We thank the reviewer for highlighting this issue and suggesting that the task description be more precise.
> However, we hope the unique aspects and challenges of the task are adequately appreciated.
>
> **[W2. The details.]**
> 1) **The Initial Point of the Denoising process.** During the training phase, the initial point consists of concatenating the target image with noise in latent space, the masked image in latent space, and the mask itself, as illustrated by z\_t' in Figure 2.
> In the inference phase, the target image with noise in latent space is replaced with Gaussian noise at the initial point.
> The above is discussed in Lines 186-189 of the paper.
> 2) **The training Objective.** FuseAnyPart uses reconstruction as a proxy task and does not require swapped paired faces.
> For instance, the target face image and the source face images, which provide different facial parts, are sampled from different images of the same identity.
> Therefore, reconstructing the target image allows the FuseAnyPart to gain the ability to generate natural images by merging different organs.
> Our paper details the training and inference pipeline in Lines 60-62.
> 3) **The dimension of the extracted feature.**
> The feature $f$ from the CLIP image encoder has a shape of $(h\times w) \times c$, which corresponds to $h\times w$ visual tokens.
> This feature $f$ can be input into an MLP to align with the dimensions of the latent features in UNet.
>
> **[W3. Questions on the CelebA dataset.]**
> The CelebA dataset now includes identity annotations in the “Anno/Identity\_CelebA.txt file,” which can be downloaded from the official CelebA website. During training with FuseAnyPart, source and target images are sampled from different images of the same identity.
> More qualitative results are added in Figure 4 in the newly submitted PDF, and they will be added in the final edition of the paper.
>
> **[W4. More comparisons.]**
> The state-of-the-art face-swapping methods supporting face parts swapping are all included in Table 1 of our paper.
> According to their papers, FuseAnyPart beats Diffswap in FID and OSim, and Diffswap beats SimSwap(2020) and FSGAN(2019).
> A qualitative comparison of FuseAnyPart with DiffFace is shown in Figure 3 of the newly submitted PDF, demonstrating a significant advantage of FuseAnyPart.

---

> > ### Comment · Reviewer_Aivf · 2024-08-12
> >
> > Thanks for this rebuttal. I raise my score to borderline accept.

---

### Official Review · Reviewer_b8uw · 2024-07-21

**Soundness:** 4
**Presentation:** 4
**Contribution:** 4
**Rating:** 9
**Confidence:** 5

**Summary:**

"FuseAnyPart: Diffusion-Driven Facial Parts Swapping via Multiple Reference Images" introduces a novel framework for swapping individual facial parts using a diffusion model that effectively utilizes multiple reference images. The paper outlines the methodological innovation and superiority of FuseAnyPart over traditional GAN-based and diffusion-based methods, which primarily focus on full-face swapping. This approach enables high-fidelity and cohesive blending of facial parts from disparate sources, enhancing fine-grained character customization capability.

**Strengths:**

*   **Originality**: This paper introduces a unique application of diffusion models to the problem of facial parts swapping, diverging from the traditional focus on full-face swaps.
*   **Quality**: Demonstrates improved quality and robustness in facial parts swapping through qualitative and quantitative results.
*   **Clarity**: The paper is articulate and well-organized, with thorough explanations and clear visual aids that enhance the understanding of the proposed method.
*   **Significance**: Offers significant practical applications in various fields, including digital media creation and personalized entertainment.

**Weaknesses:**

*   **Complexity**: The computational complexity might limit its application in real-time or on lower-end devices.
*   **Scope of Data**: More diverse testing on datasets from various demographics could enhance the robustness and generalization claims.
*   **Dependency on High-Quality Inputs**: The method's effectiveness relies heavily on the quality of input images, which could limit its applicability in less-controlled environments.

**Questions:**

1.  What is the performance impact when using lower-quality or varied lighting conditions in input images?
2.  Could this method be adapted for real-time applications, and if so, what optimizations would be necessary?
3.  How does the method perform when facial parts from significantly different racial or age groups are swapped?

**Limitations:**

The authors adequately discuss the limitations, including the high computational requirements and the dependency on high-quality reference images. They also mention potential challenges in diverse application scenarios, which is vital for setting realistic expectations for the method's deployment.

---

> ### Author Rebuttal · Authors · 2024-08-06
>
> **[W. The weakness of FuseAnyPart.]**
> 1) Diffusion models typically have high computational complexity due to the need for recursive iterations, which limits their applications to real-time or on lower-end devices.
> 2) More diverse testing results are illustrated in Figure 1 in the newly submitted PDF to validate the robustness and generalization of FuseAnyPart.
> 3) Most models rely on high-quality training datasets. The quality of images can be improved using super-resolution methods.
> For example, the dataset used in this work is CelebA-HQ, which is reconstructed with super-resolution through GANs.
>
> **[Q1. Performance of low quality inputs.]**
> The performance of FuseAnyPart may be affected by lower-quality or varying lighting conditions input images.
> However, some preprocessing techniques, such as facial alignment and **super-resolution**, can be adopted to mitigate these effects.
>
>
> **[Q2. Optimizations for real-time applications.]**
> Algorithms like **Latent Consistency Models (LCM)** can overcome the slow iterative sampling process of Diffusion Models, enabling fast inference with minimal steps instead of the usual dozens or hundreds.
> In engineering, techniques like int8 **model quantization** can significantly reduce computational load.
> Together, these strategies can speed up FuseAnyPart.
>
> **[Q3. Results of significantly different racial or age groups.]**
> These results are shown in Figure 1 in the newly submitted PDF.

---

### Author Rebuttal · Authors · 2024-08-07

Dear reviewers and meta reviewers,

We appreciate all reviewers for their valuable comments and suggestions.
We have carefully addressed the comments and added details and comparisons as follows:


- We have provided solutions to accelerate inference and improve low data quality.
- We have added a wider range of testing results, particularly for significantly different race and age groups.
- We have addressed the issue of skin color variation and provided visual demonstrations of the outcomes.
- We have detailed why the fusion mechanism poses the primary challenge in facial part swapping.
- We have clarified the training process and specific variables for better understanding.
- We have outlined the training objective of FuseAnyPart, which uses reconstruction as a proxy task.
- We have included new baseline results from DiffFace for qualitative comparison.
- We have resolved the issue concerning the identity files in the CelebA dataset.
- We have added details regarding the evaluation metric (OSim).
- We have included information on the Addition-based Injection Module.
- We have revised the term "one-step" to "simultaneously" for clarity.

We will release our code and checkpoints in the camera-ready version, and please see below our responses to each reviewer.
If you have any questions or suggestions, please feel free to leave your comments on OpenReview.

Authors of FuseAnyPart

---

### Decision · Program_Chairs · 2024-09-25

**Decision:**

Accept (spotlight)

**Comment:**

The paper received all accept recommendations after rebuttal. Reviewers found the proposed method interesting and easy to read, and their concerns have been addressed after rebuttal. AC agrees with reviewers and is happy to accept this paper for publication.